# Protective factors, risk factors, and intervention strategies in the prevention and reduction of crime among adolescents and young adults aged 12–24 years: A scoping review protocol

**Rosanna Mary Rooney**[1]*, Amber Hopkins[1], Jacob Peckover[1], Kael Coleman[1], Rebecca Sampson[1], Rosa Alati[1,2], Sharinaz Hassan[1], Christina M. Pollard[1], Jaya A. R. Dantas[1], Roanna Lobo[1], Zakia Jeemi[1], Sharyn Burns[1], Robert Cunningham[3], Stephen Monterosso[3], Lynne Millar[1,4], Sender Dovchin[5], Rhonda Oliver[5], Ranila Bhoyroo[6], Getinet Ayano[1]

**1** School of Population Health, Curtin University, Perth, WA, Australia, **2** Institute for Social Science Research, The University of Queensland, Brisbane, QLD, Australia, **3** Curtin Law School, Curtin University, Perth, WA, Australia, **4** Telethon Kids Institute, Perth, WA, Australia, **5** School of Education, Curtin University, Perth, WA, Australia, **6** Mental Health Commission, WA, Australia

* r.rooney@curtin.edu.au

## Abstract

### Background

Evidence indicates that criminal behaviour in youth is linked with a range of negative physical, mental, and social health consequences. Despite a global decrease over the last 30 years, youth crime remains prevalent. Identifying and mapping the most robust risk and protective factors, and intervention strategies for youth crime could offer important keys for predicting future offense outcomes and assist in developing effective preventive and early intervention strategies. Current reviews in the area do not include literature discussing at risk populations such as First Nations groups from countries such as Australia, Canada and New Zealand. This is a critical gap given the disproportionally high rates of incarceration and youth detention among First Nations people globally, particularly in countries with a colonial past. The aim of this scoping review is to identify and map the key risk and protective factors, along with intervention strategies, that are essential for recognizing adolescents and young adults at risk of crime.

### Methods

This scoping review protocol has been developed in line with the Arksey and O'Malley framework and the Joanna Briggs Institute (JBI) Reviewers' Manual. The review protocol was preregistered with Open Science Framework (https://osf.io/kg4q3). ProQuest, PubMed, Web of Science, Scopus, and PsycInfo were used to retrieve relevant articles. Grey literature was searched using Google searches and ProQuest dissertations

---

**Funding:** This study is funded by the Department of Industry, Science, Energy and Resources Safer Communities Fund, Australia. The funder had no further role in data collection, study design, analyses, and interpretation of the findings.

**Competing interests:** The authors have declared that no competing interests exist.

databases. Original research articles examining protective factors, risk factors, and intervention strategies for prevention and reduction of crime in 12-24-year-olds were included. Two independent reviewers conducted eligibility decisions and data extraction. Findings has been reported in accordance with the Preferred Reporting Items for Systematic reviews and Meta-Analyses extension for Scoping Reviews.

## Conclusion

Anticipated findings suggest that current research has extensively examined factors across all levels of the socioecological model, from individual to community levels, revealing a predominant focus on individual-level predictors such as substance use, prior criminal history, and moral development. The review is expected to identify effective interventions that address critical factors within each domain, including Multisystemic Therapy (MST) and Multidimensional Treatment Foster Care (MTFC), which have shown promise in reducing youth crime. Additionally, it will likely highlight significant trends in risk and protective factors, such as the dual role of academic achievement—both as a risk and protective factor—and the impact of family-based interventions. The review will also address gaps in research, particularly regarding Indigenous youth, underscoring the need for targeted studies to better understand their unique challenges. These findings will guide future research and inform the development of comprehensive prevention and early intervention programs tailored to diverse youth populations.

## Introduction

Crime among young people remains a pressing issue globally, with a particular emphasis on drug-related offenses such as possession and trafficking. While there has been a general decline in youth offending over the past three decades, especially in property-related and violent crimes, regional variations persist [1, 2]. For instance, in the United States, the number of violent crime arrests involving youth reached a new low by 2020, representing a 78 percent decrease from the 1994 peak and a 56 percent decline between 2010 and 2020 [2]. Despite this general decline, drug-related offenses have been increasing in certain regions, such as Europe. From 1990 to 2000, Western Europe saw a notable rise in drug offenses, correlated with the increased availability of drugs in European markets and the rise of drug use as shown by other indicators [1]. Factors contributing to this decline include adjustments in adolescent parenting, changes in socialisation practices, decreased youth alcohol consumption, and improved access to health services and welfare [3–6]. However, certain demographics, notably First Nations youth in countries like Australia, Canada, and New Zealand, continue to be disproportionately represented in the criminal justice system, indicating systemic issues that need addressing [7, 8].

Youth involvement in criminal behaviour carries severe physical, mental, and social health consequences, underscoring the urgent need for effective prevention and early intervention strategies. The health impacts of youth crime are profound: young offenders and victims often experience severe mental health issues, including increased rates of depression, anxiety, and trauma, which can have lasting effects on their development and overall well-being [9]. These mental health challenges are compounded by physical health problems such as injuries sustained during criminal activities and long-term health complications associated with substance

abuse [10]. The link between drug use and criminal behaviour not only exacerbates these health issues but also perpetuates a cycle of addiction and offending.

The significance of addressing youth crime extends beyond individual health concerns to broader societal impacts. Families and communities face heightened healthcare costs and diminished quality of life due to the repercussions of youth offending. Communities experience reduced safety, increased fear, and economic costs associated with criminal justice responses. Risk factors contributing to youth offending are varied and include family dysfunction, exposure to abuse or community violence, substance misuse, and socioeconomic disparities [16–20]. hese factors highlight the complexity of the issue and the need for targeted prevention strategies that address both the immediate and underlying causes of youth criminal behaviour.

Understanding these severe consequences and diverse risk factors is crucial for developing comprehensive approaches to prevent youth crime and mitigate its impact on individuals and communities [11, 12]. Contact with the justice system often exacerbates these issues, leading to diminished autonomy, mental health struggles, and increased risk of self-harm or suicide attempts [13–16]. Protective factors such as education, positive peer or community relationships, and access to trauma-informed services can mitigate these risks and foster resilience in young people [17–20].

Effective intervention strategies for youth crime encompass both therapeutic and non-therapeutic approaches. Therapeutic interventions like cognitive behavioural therapy and dialectical behavioural therapy have shown promise in addressing risk-taking behaviours and reducing offending [21–24]. Functional family therapy, grounded in a multi-systemic and trauma-informed perspective, offers a holistic approach to addressing familial dynamics [24]. Moreover, culturally safe practices, peer support, and flexibility in therapeutic approaches are crucial components of effective interventions with at-risk youth [25].

Effective intervention strategies for youth crime encompass both therapeutic and non-therapeutic approaches. Therapeutic interventions like cognitive behavioural therapy (CBT) and dialectical behavioural therapy (DBT) have shown promise in addressing risk-taking behaviours and reducing offending, yet there remains debate about their overall effectiveness and applicability to diverse populations [17, 26]. Critics, such as Malvaso et al., argue that while these therapies can be beneficial, they may not adequately address broader socio-economic and systemic factors contributing to criminal behavior [27]. For instance, the efficacy of CBT and DBT can vary significantly based on individual differences and the quality of implementation, raising concerns about their universal applicability. Additionally, Functional Family Therapy (FFT), which offers a holistic approach by addressing familial dynamics and employing a multi-systemic and trauma-informed perspective, is praised for its comprehensive approach. However, challenges such as engaging all family members and maintaining consistent implementation can impact its success [26]. The debate also extends to the need for culturally safe practices, peer support, and flexible therapeutic approaches, which are crucial for effective interventions with at-risk youth but can be challenging to implement consistently [27]. These ongoing controversies highlight the complexities and varied outcomes associated with different therapeutic strategies, underscoring the need for continued evaluation and adaptation of interventions.

## Aim and significance of study

Our preliminary review of the existing literature reveals a critical gap in meta-analyses or umbrella reviews specifically addressing risk and protective factors, as well as prevention strategies for youth crime, particularly for First Nations people who are disproportionately

incarcerated [25, 28]. The existing research predominantly focuses on broad or non-specific populations, leaving a significant void in targeted analyses for marginalized groups. Identifying robust risk and protective factors is essential for developing precise predictive tools and effective intervention strategies. This scoping review aims to address this gap by systematically mapping the key risk and protective factors, as well as the intervention strategies, to prevent and reduce crime among youth aged 12–24 who are at risk of offending.

The significance of this study lies in its potential to provide a comprehensive overview of factors influencing youth crime, tailored to the unique needs of First Nations communities and other high-risk groups. By consolidating and synthesizing existing research, this review will offer critical insights that can guide the development of targeted prevention and early intervention programs. The findings are expected to inform policy, improve program design, and contribute to reducing youth crime rates. Ultimately, this study will establish a benchmark for future research, helping to bridge existing gaps and foster more effective and culturally relevant interventions.

## Methods

The protocol design was guided by PRISMA guidelines (PRISMA extension for scoping reviews (PRISMA-ScR) [29] (S1 Fig) and the Arksey and O'Malley framework [30], which follows five steps: (i) identifying the research question; (ii) identifying relevant studies; (iii) study selection; (iv) charting the data; and (v) collating, summarising, and reporting results. The scoping review method was selected as it is useful for identifying gaps in, and the range of literature in an area [31]. This allows us to outline the knowledge gap that exists relating to intervention strategies for First Nations youth. The protocol was preregistered with the Open Science Framework (https://osf.io/vka3p).

### Step 1: Identification of the research questions

An appropriate set of research questions constitutes the foundation of a scoping review, influencing the literature explored and the data extracted [29, 30]. To gather a detailed understanding of the topic area and guide the development of the research questions, a preliminary literature review was conducted. It is recommended that research questions in a scoping review are broad and articulate the target population, concept and context [30]. Following these recommendations and considering data collected in the initial literature review, the following research questions were formulated:

1. What are the risk and protective factors influencing criminal behavior among adolescents and young adults aged 12 to 24?

2. What is the extent, range, and nature of the evidence on crime reduction and prevention strategies for adolescents and young adults aged 12 to 24?

3. Which intervention strategies are most promising for reducing and preventing crime among adolescents and young adults aged 12 to 24?

### Step 2: Identifying relevant studies

**Eligibility criteria.** As recommended by the JBI Reviewers' Manual [32, 33], the 'Population Concept Context (PCC)' framework informed the inclusion criteria. (**see Table 1** for a summary of PCC inclusion criteria). The population of interest in this scoping review is youth aged between 12–24 years of age who have engaged in criminal behaviour or are deemed as being at risk of offending globally. Crime was defined as any of the following behaviours:

**Table 1. Population, Concept and Context (PCC) inclusion criteria.**

| Population | Youth aged 12–24 years who have committed a crime |
|---|---|
| Concept | Risk factors |
| | Protective factors |
| | Intervention strategies |
| | Recidivism, persistent offending, and any other type of offending |
| Context | Global studies with observational and interventional designs |
| | Reported in the English language |
| | Grey and unpublished literature |

- Theft, robbery, and burglary

- Intentionally causing harm or death to another person

- Sexual assault and related offences

- Dangerous or negligent acts endangering persons

- Unlawful entry and break and enter offences

- Traffic and vehicle regulatory offences

- Offences against justice procedures.

Studies that examined risk factors, protective factors, and/or intervention strategies to prevent or reduce crime in the population of interest were all considered for selection. All types of offending (e.g., persistent, recidivism) were also considered for selection. Studies were either observational or interventional in design, as these appropriately aligned with the research questions. Studies from the following contexts were taken under consideration:

a. Global Context: Studies conducted across various countries and regions to provide a broad understanding of the issue.

b. Universal Interventions in the General Population: Interventions aimed at the entire population, not limited by specific characteristics, conditions, or risk factors. The term "general population" refers to individuals who are not specifically categorized by certain risk factors or conditions.

c. Prevention/Early Intervention in the General Population: Programs and strategies designed to prevent the onset of criminal behavior or address early signs of such behavior within the general population.

d. Targeted Interventions in Clinical and Population-Based Settings: Interventions specifically aimed at high-risk groups or those already engaged in offending behavior, within both clinical settings (e.g., mental health services) and broader community-based contexts.

e. Interventions for First Nations Populations: Studies focused on interventions specifically designed for Indigenous communities, recognizing the unique cultural and social contexts of these groups.

f. Grey Literature: Inclusion of non-peer-reviewed sources, such as reports, theses, and government publications, to capture a wider range of evidence.

g. No Date Restriction: Studies from any publication year were eligible, allowing for a comprehensive review of the literature

h. Language: Only studies published in English were considered, to ensure accessibility and comprehension.

**Information sources and search strategy.** The search strategy (see **Table 2**) has been developed in collaboration with an academic librarian using keywords that adhere to the outlined inclusion criteria; this process was guided by the JBI manuals [32, 33]. This search strategy was configured for each of the following databases that are to be used: ProQuest, PubMed, Web of Science, Scopus, and PsycINFO. Grey and unpublished literature were searched using Google search, and ProQuest dissertations databases. All identified literature were stored and managed through the EndNote v.20.2 software [34]. The final details regarding any adjustments to the search strategy or search dates were delineated in the published scoping review.

## Step 3: Study selection

One author performed the search, de-duplication, and collecting of literature found following the search strategy. Two authors independently screened abstracts to identify whether the necessary inclusion criteria have been met. Sources that do not meet the criteria were excluded from further review. The same two authors then reviewed the full texts of retained articles to confirm that inclusion criteria is met. Any inter-reviewer disagreements were discussed, including a third reviewer if needed, until a consensus is reached.

## Step 4: Charting the data

Two reviewers were engaged in the data charting process. One reviewer extracted the data items recommended by the JBI Manual [35] and relevant to the research questions from each included source, while the second reviewer verified the accuracy and completeness of the data entry in the spreadsheet. Prior to this, the below data items were added to a template and piloted with a sample of included records and any necessary adjustments could be made before full charting to ensure a consistent and singular approach to charting is established.

## Data items

Specific data that were extracted from eligible studies include:

a. Citation details (author/s, title, date of publication)

b. Country

c. Study design and context

d. Participant details (age, gender, sample size, Indigenous identity)

e. History of offending or risk identified

f. Exposure and outcome assessment

In the context of this scoping review, the exposure and outcome assessment process should align with the PCC (Population, Concept, Context) framework rather than the PICO (Population, Intervention, Comparison, Outcome) framework, which is typically used for systematic reviews and meta-analyses. The PCC framework is more suitable for scoping reviews as it allows for broader exploration of the available literature without the strict comparison of interventions required by PICO.

Under the PCC framework, exposure is defined in terms of the specific characteristics or conditions within the population of interest (e.g., demographic factors, socio-economic status,

**Table 2.**

| No. | Search topic | Search keywords (titles, abstracts, general keywords, and subject headings) |
|---|---|---|
| 1 | Condition/outcome of interest | (("crime"[MeSH Terms] OR "crime"[All Fields] OR "crimes"[All Fields] OR "crime s"[All Fields] OR ("recidivate"[All Fields] OR "recidivated"[All Fields] OR "recidivating"[All Fields] OR "recidivism"[MeSH Terms] OR "recidivism"[All Fields] OR "recidivisms"[All Fields]) OR ("criminals"[MeSH Terms] OR "criminals"[All Fields] OR "offender"[All Fields] OR "offenders"[All Fields] OR "offend"[All Fields] OR "offended"[All Fields] OR "offender s"[All Fields] OR "offending"[All Fields] OR "offends"[All Fields]) OR ("delinquencies"[All Fields] OR "delinquency"[All Fields] OR "delinquent"[All Fields] OR "delinquents"[All Fields]) OR (("delinquencies"[All Fields] OR "delinquency"[All Fields] OR "delinquent"[All Fields] OR "delinquents"[All Fields]) AND ("behavior"[MeSH Terms] OR "behavior"[All Fields] OR "behavioral"[All Fields] OR "behavioural"[All Fields] OR "behavior s"[All Fields] OR "behaviorally"[All Fields] OR "behaviour"[All Fields] OR "behaviourally"[All Fields] OR "behaviours"[All Fields] OR "behaviors"[All Fields] OR "pattern"[All Fields] OR "pattern s"[All Fields] OR "patternability"[All Fields] OR "patternable"[All Fields] OR "patterned"[All Fields] OR "patterning"[All Fields] OR "patternings"[All Fields] OR "patterns"[All Fields])) OR ("offense"[All Fields] OR "offenses"[All Fields] OR "offensive"[All Fields] OR "offensives"[All Fields])) |
| 2 | Population | ("juvenile"[All Fields] OR "juvenile s"[All Fields] OR "juveniles"[All Fields] OR "juvenility"[All Fields] OR ("adolescent"[MeSH Terms] OR "adolescent"[All Fields] OR "youth"[All Fields] OR "youths"[All Fields] OR "youth s"[All Fields]) OR ("adolescences"[All Fields] OR "adolescency"[All Fields] OR "adolescent"[MeSH Terms] OR "adolescent"[All Fields] OR "adolescence"[All Fields] OR "adolescents"[All Fields] OR "adolescent s"[All Fields]) OR ("young adult"[MeSH Terms] OR ("young"[All Fields] AND "adult"[All Fields]) OR "young adult"[All Fields]) OR ("child"[MeSH Terms] OR "child"[All Fields] OR "children"[All Fields] OR "child s"[All Fields] OR "children s"[All Fields] OR "childrens"[All Fields] OR "childs"[All Fields]) OR (("young"[All Fields] OR "youngs"[All Fields]) AND ("people s"[All Fields] OR "peopled"[All Fields] OR "peopling"[All Fields] OR "persons"[MeSH Terms] OR "persons"[All Fields] OR "people"[All Fields] OR "peoples"[All Fields])) OR ("adolescent"[MeSH Terms] OR "adolescent"[All Fields] OR "teen"[All Fields])) |
| 3 | Exposure/context | ("predictor"[All Fields] OR "predictors"[All Fields] OR ("risk"[MeSH Terms] OR "risk"[All Fields]) OR ("protective factors"[MeSH Terms] OR ("protective"[All Fields] AND "factors"[All Fields]) OR "protective factors"[All Fields]) OR ("risk factors"[MeSH Terms] OR ("risk"[All Fields] AND "factors"[All Fields]) OR "risk factors"[All Fields]) OR (("associability"[All Fields] OR "associational"[All Fields] OR "associative"[All Fields] OR "associatively"[All Fields] OR "associativity"[All Fields] OR "associator"[All Fields] OR "associators"[All Fields]) AND ("factor"[All Fields] OR "factor s"[All Fields] OR "factors"[All Fields])) OR ("analysis"[MeSH Subheading] OR "analysis"[All Fields] OR "determination"[All Fields] OR "determinant"[All Fields] OR "determinants"[All Fields] OR "determinate"[All Fields] OR "determinated"[All Fields] OR "determinates"[All Fields] OR "determinating"[All Fields] OR "determinations"[All Fields] OR "determine"[All Fields] OR "determined"[All Fields] OR "determines"[All Fields] OR "determining"[All Fields]) OR ("correlate"[All Fields] OR "correlated"[All Fields] OR "correlates"[All Fields] OR "correlating"[All Fields] OR "correlation"[All Fields] OR "correlation s"[All Fields] OR "correlations"[All Fields] OR "correlative"[All Fields] OR "correlatives"[All Fields]) OR ("factor"[All Fields] OR "factor s"[All Fields] OR "factors"[All Fields])) |
| 4 | Intervention strategies | ("intervention s"[All Fields] OR "interventions"[All Fields] OR "interventive"[All Fields] OR "methods"[MeSH Terms] OR "methods"[All Fields] OR "intervention"[All Fields] OR "interventional"[All Fields] OR ("methods"[MeSH Terms] OR "methods"[All Fields] OR ("intervention"[All Fields] AND "strategies"[All Fields]) OR "intervention strategies"[All Fields]) OR ("program"[All Fields] OR "program s"[All Fields] OR "programe"[All Fields] OR "programmed"[All Fields] OR "programmes"[All Fields] OR "programming"[All Fields] OR "programmability"[All Fields] OR "programmable"[All Fields] OR "programmably"[All Fields] OR "programme"[All Fields] OR "programmes"[All Fields] OR "programmed"[All Fields] OR "programmer"[All Fields] OR "programmer s"[All Fields] OR "programmers"[All Fields] OR "programmes"[All Fields] OR "programming"[All Fields] OR "programmings"[All Fields] OR "programs"[All Fields]) OR ("program"[All Fields] OR "program s"[All Fields] OR "programe"[All Fields] OR "programmed"[All Fields] OR "programes"[All Fields] OR "programing"[All Fields] OR "programmability"[All Fields] OR "programmable"[All Fields] OR "programmably"[All Fields] OR "programme"[All Fields] OR "programme s"[All Fields] OR "programmed"[All Fields] OR "programmer"[All Fields] OR "programmer s"[All Fields] OR "programmers"[All Fields] OR "programmes"[All Fields] OR "programming"[All Fields] OR "programmings"[All Fields] OR "programs"[All Fields]) OR ("program"[All Fields] OR "program s"[All Fields] OR "programe"[All Fields] OR "programed"[All Fields] OR "programes"[All Fields] OR "programing"[All Fields] OR "programmability"[All Fields] OR "programmable"[All Fields] OR "programmably"[All Fields] OR "programme"[All Fields] OR "programme s"[All Fields] OR "programmed"[All Fields] OR "programmer"[All Fields] OR "programmer s"[All Fields] OR "programmers"[All Fields] OR "programmes"[All Fields] OR "programming"[All Fields] OR "programmings"[All Fields] OR "programs"[All Fields]) |
| 5 | Prevent or reduce | ("reduce"[All Fields] OR "reduced"[All Fields] OR "reduces"[All Fields] OR "reducing"[All Fields] OR ("prevent"[All Fields] OR "preventability"[All Fields] OR "preventable"[All Fields] OR "preventative"[All Fields] OR "preventatively"[All Fields] OR "preventatives"[All Fields] OR "prevented"[All Fields] OR "preventing"[All Fields] OR "prevention and control"[MeSH Subheading] OR ("prevention"[All Fields] AND "control"[All Fields]) OR "prevention and control"[All Fields] OR "prevention"[All Fields] OR "prevention s"[All Fields] OR "preventions"[All Fields] OR "preventive"[All Fields] OR "preventively"[All Fields] OR "preventives"[All Fields] OR "prevents"[All Fields]) |
| Final search query | The intersection of five topics | 1 AND 2 AND 3 AND 4 AND 5 |

or environmental factors) that may influence the risk of engaging in criminal behavior. Outcomes refer to the range of consequences or effects observed within the population, including rates of offending, recidivism, or the success of intervention strategies.

Given the nature of a scoping review, the emphasis is placed on mapping and summarizing the existing evidence, identifying gaps in the literature, and understanding the context in which these exposures and outcomes are observed. Therefore, authors should ensure that the assessment process remains consistent with the PCC framework by focusing on the conceptual relationships and broader context of the findings rather than strictly comparing interventions and their outcomes.

g. Measure of association

h. Confounders adjusted for

i. Other characteristics relevant to the research questions

## Step 5: Collating, summarising, and reporting the results

The finding has been reported in accordance with the Preferred Reporting Items for Systematic Reviews and Meta-Analyses extension for Scoping Reviews [36] and the flow chart of screening and data identification processes was presented using PRISMA-ScR (**Fig 1**). The results of the scoping review were organised and presented using a flow diagram, table, and text. A flow diagram has been used to visualise the implementation of the search strategy and the process of study selection, with accompanying text to provide descriptive statistics on the number of studies in each stage. In our preliminary systematic search of electronic databases and manual search of evidence, we found 9,769 sources, of which 235 met the predefined criteria (**Fig 1**). A table was used to summarise study characteristics and provide a brief overview of the findings for each included study. Text included a narrative synthesis of the review findings, with a specific focus on the research questions addressed. A discussion of identified gaps in the research was included, as well as general limitations noted across the included studies. The socio-ecological framework specified in McLeroy et al. (1988) was used as a guide to characterise the existing information and classify the types of risk factors [37].

## Critical appraisal of individual sources of evidence

As suggested by the Arksey and O'Malley framework (2005) [30], this scoping review does not include a formal risk of bias or quality assessment for the included studies [30]. This approach is justified by the specific objectives of a scoping review, which are to map the existing literature, identify key concepts, and determine the scope of research on a given topic, rather than to evaluate the methodological quality of individual studies.

Scoping reviews are designed to provide a comprehensive overview of the breadth of research rather than in-depth evaluations of study quality. This focus on breadth aligns with the aim of the scoping review to capture a wide range of literature and identify gaps in research, which is particularly important when exploring diverse and evolving fields [31].

Moreover, the research questions formulated for this review are concerned with the scope and nature of evidence related to risk factors, protective factors, and intervention strategies rather than the quality of the studies themselves. By including a broad spectrum of studies, the review seeks to offer an extensive overview of the evidence available without imposing restrictive criteria that could limit the inclusion of potentially relevant research [32].

Thus, the decision to omit a risk of bias or quality assessment is in line with the methodological approach of scoping reviews, which prioritize inclusivity and comprehensiveness over

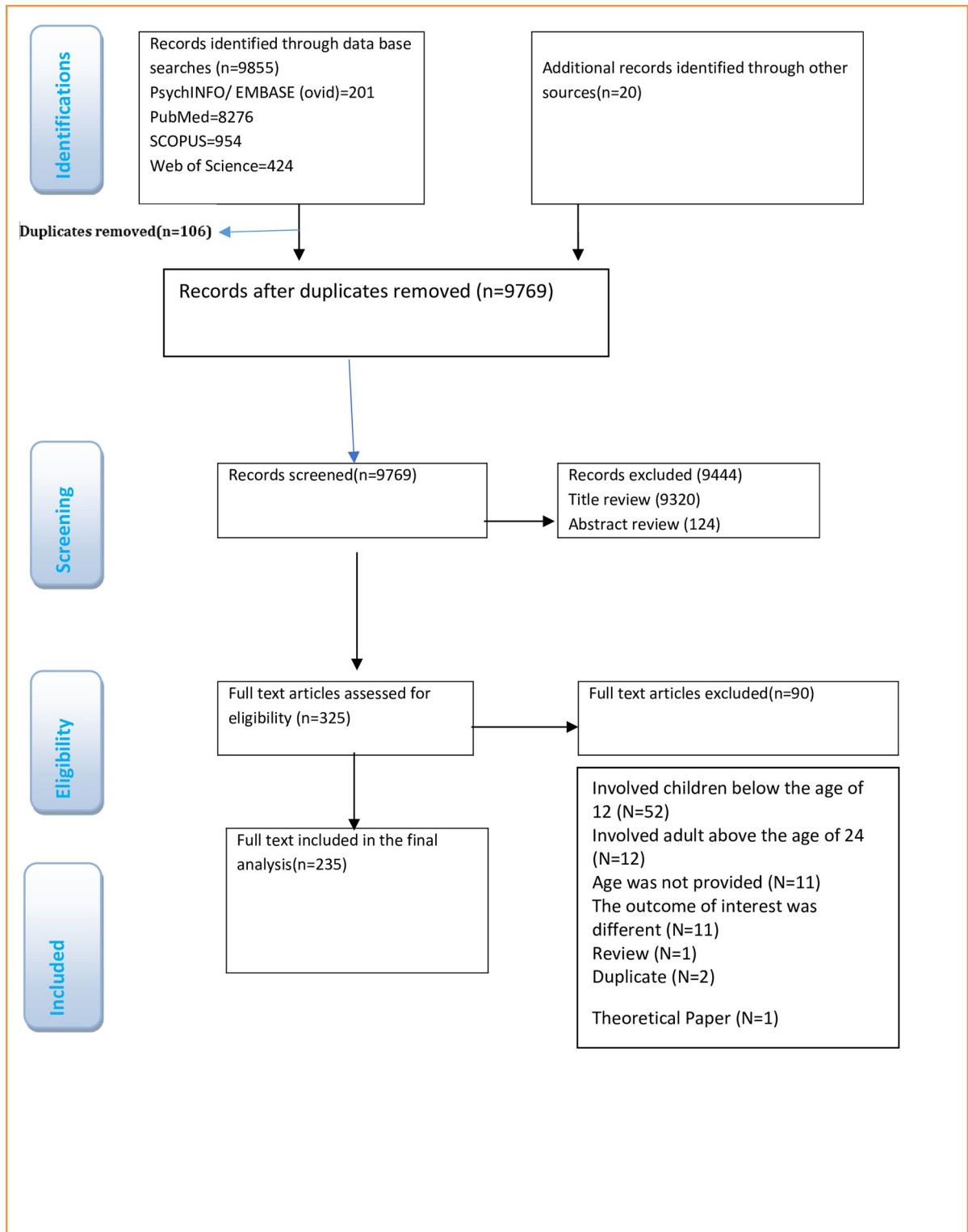

**Fig 1. Flow diagram for the literature review, which illustrates the selection criteria and the flow of information through the different phases of the review, including database searches and the number of studies in each phase.**

critical appraisal. This approach is supported by the framework and methodology outlined by Levac et al., which underscores the importance of capturing the breadth of research to inform future studies and identify areas requiring further investigation [31].

## Discussion

Given the high degree of consequence that surrounds youth crime globally, it is critical to review factors that can both prevent criminal behaviour in youth through addressing key risk and protective factors and early intervention strategies. It is particularly concerning that despite disproportionate rates of incarceration of First Nations youth, especially in countries with a colonial past, there have been no meta-analyses or umbrella reviews of risk, protective or intervention strategies in First Nations youth aimed at preventing or reducing crime globally. In the absence of these reviews, it is therefore essential to perform a scoping review of all relevant grey literature, published and unpublished research in order to evidence of what is working. The synthesis of these results contributed towards the planning of intervention strategies that can be used to prevent and reduce crime in at risk populations. The findings of the scoping review will be disseminated through publication in high impact peer reviewed journals and communicated to relevant services, organisations and communities through reports, community meetings, social media channels, and newsletters.

In our preliminary analyses, we identified several critical risk factors that significantly contribute to youth crime. Among these, substance use emerged as a predominant predictor, highlighting the role of addiction and substance abuse in escalating criminal behavior [38–40]. Lower academic achievement was also identified as a key risk factor, where youth with poor academic performance were more likely to engage in criminal activities, underscoring the importance of educational support and interventions [38–40]. Poor parental supervision was another significant factor, with inadequate parental oversight often leading to increased risk of youth involvement in crime [38–40]. Additionally, social disadvantage, including factors such as low income, unemployment, and living in disadvantaged neighborhoods, was strongly associated with higher levels of criminal behavior among youth [38–40]. To mitigate these risks, our analysis revealed two particularly effective protective interventions. The first is multisystemic therapy (MST), an evidence-based approach that targets the various interconnected factors contributing to delinquent behavior [41]. MST involves intensive interventions at the family and community levels, aiming to strengthen family dynamics, improve communication, and reduce the influence of negative peer groups. Studies consistently show that youth who undergo MST exhibit significantly lower levels of criminal activity [41].

The second intervention is Multidimensional Treatment Foster Care (MTFC), which provides specialized foster care combined with comprehensive therapeutic interventions for at-risk youth [42]. MTFC is designed to address behavioral issues through structured support, including close supervision, consistent discipline, and positive reinforcement within a foster care setting. Research indicates that MTFC effectively reduces delinquent behavior and improves overall outcomes for youth involved in the program [42].

These findings suggest that addressing the identified risk factors through targeted interventions like MST and MTFC can play a crucial role in reducing youth crime and supporting at-risk individuals in developing healthier, more positive life trajectories [41, 42].

### Limitations

A potential limitation of the scoping review is that there may be a large volume of literature which may result in limited depth or review for each individual study. In addition, a further limitation is that as all types of literature were considered, it may be considered less rigorous

than a systematic review, meta-analysis, or umbrella review. This may limit the impact on policy and practice, however, it is an important starting point in providing an overview of the current risk, protective and intervention strategies that have been used to prevent and reduce offending in youth at risk of crime including First Nations populations.

## Supporting information

**S1 Fig. PRISMA checklist for the flow diagram of the literature review.**
(PDF)

## Author Contributions

**Conceptualization:** Rosanna Mary Rooney, Getinet Ayano.

**Data curation:** Getinet Ayano.

**Funding acquisition:** Rosa Alati, Christina M. Pollard, Jaya A. R. Dantas, Roanna Lobo, Sharyn Burns, Robert Cunningham, Stephen Monterosso, Lynne Millar, Sender Dovchin.

**Investigation:** Rosa Alati, Christina M. Pollard, Roanna Lobo, Sharyn Burns, Stephen Monterosso, Sender Dovchin, Rhonda Oliver, Ranila Bhoyroo.

**Methodology:** Rosanna Mary Rooney, Rebecca Sampson, Getinet Ayano.

**Project administration:** Rebecca Sampson.

**Resources:** Zakia Jeemi, Robert Cunningham, Stephen Monterosso, Lynne Millar.

**Software:** Getinet Ayano.

**Supervision:** Rhonda Oliver, Getinet Ayano.

**Validation:** Rosanna Mary Rooney, Sharinaz Hassan, Jaya A. R. Dantas, Stephen Monterosso, Getinet Ayano.

**Visualization:** Rosanna Mary Rooney, Christina M. Pollard, Getinet Ayano.

**Writing – original draft:** Rosanna Mary Rooney, Amber Hopkins, Jacob Peckover, Kael Coleman, Getinet Ayano.

**Writing – review & editing:** Rosanna Mary Rooney, Amber Hopkins, Jacob Peckover, Kael Coleman, Rebecca Sampson, Rosa Alati, Sharinaz Hassan, Christina M. Pollard, Jaya A. R. Dantas, Roanna Lobo, Zakia Jeemi, Sharyn Burns, Robert Cunningham, Stephen Monterosso, Lynne Millar, Sender Dovchin, Rhonda Oliver, Ranila Bhoyroo, Getinet Ayano.

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
