## [Decision Letter · Decision Letter 0]

19 Feb 2024

PONE-D-23-15234Protective factors, risk factors, and intervention strategies in the prevention and reduction of crime in 12–24-year-old: A Scoping Review ProtocolPLOS ONE

Dear Dr. Rooney, 

Thank you for submitting your manuscript to PLOS ONE. After careful consideration, we feel that it has merit but does not fully meet PLOS ONE’s publication criteria as it currently stands. Therefore, we invite you to submit a revised version of the manuscript that addresses the points raised during the review process.

Please submit your revised manuscript by  Apr 04 2024 11:59PM**.** If you will need more time than this to complete your revisions, please reply to this message or contact the journal office at plosone@plos.org. Please include the following items when submitting your revised manuscript:A rebuttal letter that responds to each point raised by the academic editor and reviewer(s). You should upload this letter as a separate file labeled 'Response to Reviewers'.A marked-up copy of your manuscript that highlights changes made to the original version. You should upload this as a separate file labeled 'Revised Manuscript with Track Changes'.An unmarked version of your revised paper without tracked changes. You should upload this as a separate file labeled 'Manuscript'.

We look forward to receiving your revised manuscript.

Kind regards,

Johannes John-Langba, Ph.D., M.P.H., M.S.W

Academic Editor

PLOS ONE

Journal Requirements:

Additional Editor Comments:

Both reviewers however think that the manuscript is presented in an intelligible fashion and written in standard English. But although one of the reviewers evaluates the manuscript as technically sound with data that supports the conclusions,  both reviewers howefer reported that all data underlying the findings in their manuscript have not been made fully available.  

You are therefore invited to undertake a major revision of the manuscript and submit a revised version of the manuscript that addresses the points raised by both reviwers.

Reviewers' comments:

Reviewer's Responses to Questions

**Comments to the Author**

1. Is the manuscript technically sound, and do the data support the conclusions?

Reviewer #1: No

Reviewer #2: Yes

2. Has the statistical analysis been performed appropriately and rigorously? 

Reviewer #1: No

Reviewer #2: N/A

3. Have the authors made all data underlying the findings in their manuscript fully available?

Reviewer #1: No

Reviewer #2: No

4. Is the manuscript presented in an intelligible fashion and written in standard English?

Reviewer #1: Yes

Reviewer #2: Yes

5. Review Comments to the Author

Reviewer #1: Based on the Abstract/Introduction, the study (Systematic Review) is an important topic. However, the manuscript is a bit confusing, overall.

1. Pages 1-3 are duplicated within the manuscript. Also, there is no clear Abstract, as the Abstract and Introduction are combined.

2. On the bottom of page 6, last paragraph, there needs to be a subheading, "Aim of Study" or "Purpose of Study."

3. Throughout the manuscript, there are both present and future tenses. If the study/review has already been conducted, it should be past tense.

4. There is no Findings section. What did you researchers find/conclude within the study/review? What do the Findings suggest in relationship to any proposed questions?

5. The last paragraph discusses limitations, but what are the limitations based on?

Based on what was presented, I do not recommend this manuscript for publication. However, through major revisions it could be and resubmitted and reviewed.

Reviewer #2: Reviewer comments: Protective factors, risk factors, and intervention strategies in the prevention and reduction of crime in 12–24-year-old: A Scoping Review Protocol

Identifying risk factors and developing risk typologies is an important area of study in crime prevention and reduction among youth. The authors have presented the gaps in the literature on at risk populations such as First Nations groups from countries such as Australia, Canada and New Zealand. As such, this paper presents the first scoping review of primary studies of protective factors, risk factors, and intervention strategies in the prevention and reduction of crime in 12-24-year-olds, inclusive of First Nations communities. The paper is well written and is addressing a very important topic, given that crime among young people as young as 12 has child protection implications and in general has social and economic consequences. However, paper has some important limitations especially regarding the method mentioned below.

Please find my comments following my review.

Abstract

1. Authors to follow the guidelines on writing an unstructured abstract in PLOS One

2. Provide a concise description of the main aim/objective of the scoping review and state this as an aim. E.g. The aim of this scoping review is…

Introduction

The introduction section can be synthesised into four succinct paragraphs

1. Paragraph one introduces the main problem

i. Youth offending and why it is a problem?

ii. What have other countries found about youth offending, risk factors and protective factors ?

iii. How have other countries dealt with youth offending and how has the identification of protective and risk factors influenced youth offending interventions?

2. Paragraph two looks at the specific problem from a contextual perspective (i.e. among First Nations communities)

i. What is the prevalence of youth offending among First Nations communities?

3. Paragraph three introduces the gap in the literature

4. Paragraph four explain/justify why the review questions/objectives lend themselves to a scoping review approach?

Methods

1. Authors to follow the PLOS One (PRISMA-ScR) Checklist on structuring the methods section

a. Protocol and registration

b. Eligibility criteria

c. Information sources

d. Search

e. Selection of sources of evidence

f. Data charting process

g. Data items

h. Critical appraisal of individual sources of evidence

i. Synthesis of results

2. Outline what the five steps of the Arksey and O’Malley framework are

3. What framework was used to inform the development of the research questions?

4. Clarify if the data extraction process is informed by the Joanna Briggs Institute’s template for data extraction or which tool will be used for this.

6. PLOS authors have the option to publish the peer review history of their article (what does this mean?). If published, this will include your full peer review and any attached files.

Reviewer #1: No

Reviewer #2: No

---

## [Author Response · Author response to Decision Letter 0]

2 May 2024

Attached as a point-by-point response letter.

---

## [Decision Letter · Decision Letter 1]

19 Aug 2024

PONE-D-23-15234R1Protective factors, risk factors, and intervention strategies in the prevention and reduction of crime in 12–24-year-old: A Scoping Review ProtocolPLOS ONE

Dear Dr. Rooney,

Thank you for submitting your manuscript to PLOS ONE. After careful consideration, we feel that it has merit but does not fully meet PLOS ONE’s publication criteria as it currently stands. Therefore, we invite you to submit a revised version of the manuscript that addresses the points raised during the review process.

Please submit your revised manuscript by Oct 03 2024 11:59PM. If you will need more time to complete your revisions, please reply to this message or contact the journal office at plosone@plos.org. Please include the following items when submitting your revised manuscript:A rebuttal letter that responds to each point raised by the academic editor and reviewer(s). You should upload this letter as a separate file labeled 'Response to Reviewers'.A marked-up copy of your manuscript that highlights changes made to the original version. You should upload this as a separate file labeled 'Revised Manuscript with Track Changes'.An unmarked version of your revised paper without tracked changes. You should upload this as a separate file labeled 'Manuscript'.If applicable, we recommend that you deposit your laboratory protocols in protocols.io to enhance the reproducibility of your results. Protocols.io assigns your protocol its own identifier (DOI) so that it can be cited independently in the future. For instructions see: https://journals.plos.org/plosone/s/submission-guidelines#loc-laboratory-protocols. Additionally, PLOS ONE offers an option for publishing peer-reviewed Lab Protocol articles, which describe protocols hosted on protocols.io. Read more information on sharing protocols at https://plos.org/protocols?utm_medium=editorial-email&utm_source=authorletters&utm_campaign=protocols.

We look forward to receiving your revised manuscript.

Kind regards,

AKM Alamgir, PhD

Academic Editor

PLOS ONE

Journal Requirements:

Reviewers' comments:

Reviewer's Responses to Questions

**Comments to the Author**

1. If the authors have adequately addressed your comments raised in a previous round of review and you feel that this manuscript is now acceptable for publication, you may indicate that here to bypass the “Comments to the Author” section, enter your conflict of interest statement in the “Confidential to Editor” section, and submit your "Accept" recommendation.

Reviewer #3: (No Response)

Reviewer #4: (No Response)

2. Is the manuscript technically sound, and do the data support the conclusions?

Reviewer #3: No

Reviewer #4: Yes

3. Has the statistical analysis been performed appropriately and rigorously? 

Reviewer #3: N/A

Reviewer #4: N/A

4. Have the authors made all data underlying the findings in their manuscript fully available?

Reviewer #3: No

Reviewer #4: No

5. Is the manuscript presented in an intelligible fashion and written in standard English?

Reviewer #3: Yes

Reviewer #4: Yes

6. Review Comments to the Author

Reviewer #3: (No Response)

Reviewer #4: 2. This manuscript presents a well-organized and methodologically sound scoping review protocol. The authors have used established frameworks, such as the Arksey and O'Malley framework and PRISMA-ScR guidelines which are appropriate for the study. While this is a protocol and does not yet include data or conclusions, the outlined plan is robust and should effectively guide the research process.

3. Given that this is a scoping review protocol, statistical analysis is not conducted. The manuscript is focused on mapping existing literature, identify gaps and provide an overview of key concepts and evidence.

4. Since this is a protocol, no findings or data are presented yet. Once the scoping review is completed, the authors will need to ensure that all data underlying their results are made fully available in accordance to PLOS's data availability requirement.

5. The manuscript is clearly written, well structured and easy to follow. The authors have effectively communicated their research plan, and the language used is precise and professional.

General comments: Overall, this manuscript outlines a clear and comprehensive protocol for conducting a scoping review on youth crime, with a particular emphasis on identifying key risk and protective factors as well as intervention strategies, especially for First Nations Populations. The methodology is appropriate, and the writing is clear.

7. PLOS authors have the option to publish the peer review history of their article (what does this mean?). If published, this will include your full peer review and any attached files.

Reviewer #3: No

Reviewer #4: **Yes: **Ms. Susan Mary Pradhan

---

## [Author Response · Author response to Decision Letter 1]

11 Sep 2024

Attached as a point-by-point response letter.

---

## [Decision Letter · Decision Letter 2]

6 Oct 2024

PONE-D-23-15234R2Protective factors, risk factors, and intervention Strategies in the prevention and reduction of crime among adolescents and young adults aged 12-24 years: A scoping review protocolPLOS ONE

Dear Dr. Rooney,

Thank you for submitting your manuscript to PLOS ONE. After careful consideration, we feel that it has merit but does not fully meet PLOS ONE’s publication criteria as it currently stands. Therefore, we invite you to submit a revised version of the manuscript that addresses the points raised during the review process.

We look forward to receiving your revised manuscript.

Kind regards,

AKM Alamgir, PhD

Academic Editor

PLOS ONE

Journal Requirements:

Reviewers' comments:

Reviewer's Responses to Questions

**Comments to the Author**

1. If the authors have adequately addressed your comments raised in a previous round of review and you feel that this manuscript is now acceptable for publication, you may indicate that here to bypass the “Comments to the Author” section, enter your conflict of interest statement in the “Confidential to Editor” section, and submit your "Accept" recommendation.

Reviewer #3: All comments have been addressed

2. Is the manuscript technically sound, and do the data support the conclusions?

Reviewer #3: Partly

3. Has the statistical analysis been performed appropriately and rigorously? 

Reviewer #3: N/A

4. Have the authors made all data underlying the findings in their manuscript fully available?

Reviewer #3: Yes

5. Is the manuscript presented in an intelligible fashion and written in standard English?

Reviewer #3: Yes

6. Review Comments to the Author

Reviewer #3: Dear Authors,

I would like to commend you all for the effort in improving your review protocol manuscript and appreciate the opportunity to review your impeccable work.

Please see the minor revisions outlined below:

Abstract:

• Conclusion (Page 2, lines 63-65): Remove the sentence: “This scoping review protocol aims to systematically map key risk and protective factors, as well as intervention strategies, for preventing and reducing youth crime, including among First Nations populations.”

• Conclusion (revision): Remove the dash in “model—from” throughout the paragraph, as it does not reflect a professional writing style:

“Anticipated findings suggest that current research has extensively examined factors across all levels of the socioecological model, from individual to community levels, revealing a predominant focus on individual-level predictors such as substance use, prior criminal history, and moral development. The review is expected to identify effective interventions that address critical factors within each domain, including Multisystemic Therapy (MST) and Multidimensional Treatment Foster Care (MTFC), which have shown promise in reducing youth crime. Additionally, it will likely highlight significant trends in risk and protective factors, such as the dual role of academic achievement—both as a risk and protective factor—and the impact of family-based interventions. The review will also address gaps in research, particularly regarding Indigenous youth, underscoring the need for targeted studies to better understand their unique challenges. These findings will guide future research and inform the development of comprehensive prevention and early intervention programs tailored to diverse youth populations.”

• Ensure that the abstract does not exceed 300 words, as per the journal's guidelines:

Introduction:

• Page 5 (Introduction, line 129): Include the author’s surname in the sentence: “Critics argue that…”.

Critical Appraisal of Individual Sources of Evidence:

• Revise the structure of the following sentence (Page 12, Line 291): “In accordance with the Arksey and O’Malley framework (2005) [30], the scoping review does not involve a formal risk of bias or quality assessment of included sources.”

• Replace the term “Forgo” in the sentence on Page 12, line 305: “Thus, the decision to forgo a risk of bias….” with a more alternative term that will accommodate non-academic audience.

• Adhere to the journal’s in-text referencing guidelines: “This approach is supported by the framework and methodology outlined by Levac et al. (2010)...”

PRISMA Flowchart:

• (Page 19, lines 475-476): In the flowchart, add a category for duplicates removed during the IDENTIFICATION phase to confirm whether additional records identified were included in the total screened records.

General:

• The manuscript outlines a clear and robust protocol for conducting a scoping review on this topical area. Once the analysis is completed, ensure that the results are made accessible in appropriate repositories.

7. PLOS authors have the option to publish the peer review history of their article (what does this mean?). If published, this will include your full peer review and any attached files.

Reviewer #3: No

---

## [Author Response · Author response to Decision Letter 2]

9 Oct 2024

Attached as point-point response letter.

---

## [Editor Report · Decision Letter 3]

11 Oct 2024

Protective factors, risk factors, and intervention Strategies in the prevention and reduction of crime among adolescents and young adults aged 12-24 years: A scoping review protocol

PONE-D-23-15234R3

Dear Dr. Rooney,

We’re pleased to inform you that your manuscript has been judged scientifically suitable for publication and will be formally accepted for publication once it meets all outstanding technical requirements.

Kind regards,

AKM Alamgir, PhD

Academic Editor

PLOS ONE
---

## [Editor Report · Acceptance letter]

8 Nov 2024

PONE-D-23-15234R3 

PLOS ONE

Dear Dr. Rooney, 

I'm pleased to inform you that your manuscript has been deemed suitable for publication in PLOS ONE. Congratulations! Your manuscript is now being handed over to our production team.

Kind regards, 

on behalf of

Dr AKM Alamgir 

Academic Editor

PLOS ONE